# Revisiting Active Sets for Gaussian Process Decoders

**Pablo Moreno-Muñoz**[*]    **Cilie W. Feldager**[*]    **Søren Hauberg**
Section for Cognitive Systems
Technical University of Denmark (DTU)
{pabmo,cife,sohau}@dtu.dk

## Abstract

Decoders built on Gaussian processes (GPs) are enticing due to the marginalisation over the non-linear function space. Such models (also known as GP-LVMs) are often expensive and notoriously difficult to train in practice, but can be scaled using variational inference and inducing points. In this paper, we revisit active set approximations. We develop a new stochastic estimate of the log-marginal likelihood based on recently discovered links to cross-validation, and we propose a computationally efficient approximation thereof. We demonstrate that the resulting stochastic active sets (SAS) approximation significantly improves the robustness of GP decoder training, while reducing computational cost. The SAS-GP obtains more structure in the latent space, scales to many datapoints, and learns better representations than variational autoencoders, which is rarely the case for GP decoders.

## 1   Introduction

Generative models can be viewed as regression models from unknown inputs. That is, we assume $x = f(z)$, where $f$ is an unknown mapping from latent variables $z$ to observations $x$. Given the inherent difficulty of this task, it is perhaps sensible to marginalize the unknown mapping $f$ to avoid the brittleness of point estimates. This is the driving idea in the *Gaussian process latent variable* (GP-LVM) (Lawrence, 2005), which places a Gaussian process (GP) prior over the unknown mapping $f$ and marginalizes accordingly. This contrasts contemporary generative models that predominantly operate with point-estimates of $f$ (Kingma and Welling, 2013; Sohl-Dickstein et al., 2015). While conceptually elegant, the GP-LVM is, however, notoriously difficult to train and the conceptual benefits are often not realized in practice.

Exact inference involves computing the marginal likelihood, but (like other GP methods) its cubic complexity in the number of observations $\mathcal{O}(N^3)$ limits the scalability of the GP-LVM. However, the idea of marginalizing the decoder is sufficiently attractive to motivate the development of scalable and reliable training techniques: Its Bayesian formulation (Titsias and Lawrence, 2010) variationally integrates out the latent variables $z$ to obtain an evidence lower bound. Using auxiliary inducing variables, Snelson and Ghahramani (2006) expanded GP regression, which is also applicable in the unsupervised learning setting (i.e. the GP-LVM).

However, considering inducing variables here involves dangers. First, the convergence of inducing points is well-studied in the supervised GP scenario, where *inputs are fixed*, but it differs from the unsupervised case where the *inputs are estimated*. Bauer et al. (2016) notes that even in the supervised setting, inducing points are "*not completely trivial to optimise, and often tricks [...] are required*", and we hypothesize that this is further complicated in the unsupervised setting where we optimize both latent and inducing variables while they interact.

**In this paper**, we revisit *active sets* for scaling GP decoders, a sparse approximation predominantly used before the seminal work of Snelson and Ghahramani (2006). From a practical viewpoint, active

---

[*]Equal contribution.

36th Conference on Neural Information Processing Systems (NeurIPS 2022).

sets are fixed inducing variables that belong to the training dataset. We make links between such active sets and cross-validation, allowing us to lean on a recent result from Fong and Holmes (2020) which, in turn, links cross validation and the log-marginal likelihood. We show how these links allow for a stochastic estimate of the log-marginal likelihood, and that active sets can be seen as a computationally efficient approximation of this. Practically, this amounts to repeatedly and randomly sampling active sets rather than trying to find the optimal active set. We denote this framework as *stochastic active sets (SAS)*. We demonstrate that SAS consistently results in significantly better-fitted GP decoders over models trained using inducing points.

**Historical remarks.** Methods based on subsets of data diminish the computational demand and were first introduced in a GP context in the foundational work on sparse approximations by Smola and Bartlett (2001). Back then, Quiñonero-Candela and Rasmussen (2005) had already pointed out the main difficulties behind the optimal selection of subsets:

> *"Traditionally, sparse models have very often been built upon a carefully chosen subset of the training inputs. [...] In sparse Gaussian processes it has also been suggested to select the inducing inputs* $\mathbf{X_u}$ *from among the training inputs. Since this involves a prohibitive combinatorial optimization, greedy optimization approaches have been suggested [...]. Recently, Snelson and Ghahramani (2006) have proposed to relax the constraint that the inducing variables must be a subset of training/test cases, turning the discrete selection problem into one of continuous optimization."*

This explains how inducing variables reshaped the Gaussian process community, effectively banishing other subset-based methods. Our work builds on the advances made in stochastic optimization in the time since active sets were left behind. We show a third way over those considered by Quiñonero-Candela and Rasmussen (2005): instead of *optimizing* the active set, we *average* with respect to it. This simplifies matters notably and makes them more robust.

To justify our approach, we establish a link between active sets and cross validation (CV). The latter has a long history for model selection in GPs, dating at least to the seminal work of Wahba (1990). For probabilistic models, Rasmussen and Williams (2006) point to the utility of CV variants within negative log-probabilities. Building on results from Fong and Holmes (2020) linking CV and log-marginal likelihoods, we argue that, for GP-LVMs, active sets combine more gracefully with stochastic optimization. The remainder of this paper elaborates on this viewpoint and demonstrates it empirically.

## 2   Gaussian Process Decoders

The Gaussian process latent variable model (GP-LVM) (Lawrence, 2005) defines a *decoder* which is a non-linear mapping[2] $\boldsymbol{x} = f(\boldsymbol{z})$ from the latent space $\mathcal{Z} \in \mathbb{R}^Q$ to observation space $\mathcal{X} \in \mathbb{R}^D$. The prior on this map is a Gaussian process (GP) so it is drawn like $f \sim \mathcal{GP}(0, k_{\boldsymbol{\theta}}(\cdot, \cdot))$, where $k_{\boldsymbol{\theta}}$ is the covariance function or kernel and $\boldsymbol{K}_{NN}$ denotes the evaluated kernel function so the $i, j$th element of $\boldsymbol{K}_{NN}$ equals $k_{\boldsymbol{\theta}}(z_i, z_j)$. For clarity, we omit the dependence on covariance function parameters, $\boldsymbol{\theta}$.

The original version of the GP-LVM starts from one-dimensional observations $\boldsymbol{x} = \{\boldsymbol{x}_n\}_{n=1}^N$ and latent variables $\boldsymbol{z} = \{\boldsymbol{z}_n\}_{n=1}^N$, and factorises the joint distribution of the model as $p(\boldsymbol{x}, f | \boldsymbol{z}) = p(\boldsymbol{x} | f, \boldsymbol{z}) p(f | \boldsymbol{z})$. Here the conditional distributions correspond to the likelihood model and the prior

$$p(\boldsymbol{x}|f, \boldsymbol{z}) = \prod_{n=1}^N \mathcal{N}(\boldsymbol{x}_n | f(\boldsymbol{z}_n), \sigma^2), \qquad p(f|\boldsymbol{z}) = \mathcal{N}(f(\boldsymbol{z})|0, \boldsymbol{K}_{NN}). \qquad (1)$$

When the data dimensionality is $D > 1$, the model factorises across dimensions, and we have different mappings $f$ per $d^{\text{th}}$ feature. One of the principal assumptions of the GP-LVM is that the prior $p(f)$ regularises the smoothness of all mappings equally, so we only consider one *kernel*. This assumption can be relaxed if needed, but at increased computational cost and with more learnable hyperparameters.

---

[2]We use $\{\boldsymbol{x}, \boldsymbol{z}\}$ to denote observations and latent variables respectively, since we do not consider inducing variables. Notice that Lawrence (2005) use the notation $\{\boldsymbol{y}, \boldsymbol{x}\}$.

**Mapping marginalization.** A GP prior over the non-linear decoder $f$ allows for marginalisation of $f$ to obtain a closed-form expression of the marginal likelihood of the GP-LVM

$$p(\boldsymbol{x}|\boldsymbol{z}) = \int p(\boldsymbol{x}|f,\boldsymbol{z})p(f|\boldsymbol{z})\mathrm{d}f = \mathcal{N}(\boldsymbol{x}|0, \boldsymbol{K}_{NN} + \sigma^2 \mathbb{I}).$$

On a $\log$-scale, this gives the following objective function (Lawrence, 2004), which can be optimized w.r.t. both hyperparameters $\boldsymbol{\theta}$ and latent variables $\boldsymbol{z}$

$$\mathcal{L} = -\frac{DN}{2}\log 2\pi - \frac{D}{2}\log|\boldsymbol{K}_{NN} + \sigma^2 \mathbb{I}| - \frac{1}{2}\mathrm{tr}\big((\boldsymbol{K}_{NN} + \sigma^2 \mathbb{I})^{-1}\boldsymbol{x}\boldsymbol{x}^\top\big). \tag{2}$$

The difficulties of training the GP-LVM using this objective function are evident above, as the evaluation cost grows cubically with the number of observations $N$. Furthermore, notice that observations $\boldsymbol{x}$ are no longer independent (in contrast with Eq. 1) once $f$ is integrated out. The log-marginal likelihood will be the starting point for our approach in Sec. 3.

**Bayesian extension.** In the seminal works of Lawrence (2004, 2005), the GP-LVM is first derived as a non-linear extension of probabilistic principal component analysis (PPCA) (Tipping and Bishop, 1999). Considering the *isotropic* prior on the latent variables $\boldsymbol{z}$, such that $p(\boldsymbol{z}_n) = \mathcal{N}(\boldsymbol{z}_n|0, \mathbb{I}) \ \forall \boldsymbol{z}_n \in \boldsymbol{z}$, the general idea is to optimize them rather than introducing marginalization. From a *full* probabilistic perspective, one could also be interested in the posterior distribution over $\boldsymbol{z}$, which leads to using Bayesian inference for the GP-LVM approach. This is the driving idea of Titsias and Lawrence (2010), where variational methods are introduced. In particular, latent variables are not easy to marginalize, mainly due to their presence in the kernel mappings, so a lower-bound on the log-marginal likelihood $\log p(\boldsymbol{x}) = \log \int p(\boldsymbol{x}|\boldsymbol{z})p(\boldsymbol{z})\mathrm{d}\boldsymbol{z}$ of the model is derived.

So far, the Bayesian GP-LVM model (Titsias and Lawrence, 2010) has been considered as the standard methodology to apply GPs to large unsupervised datasets with $10^4 - 10^6$ observations, e.g. in regression (Bui and Turner, 2015), classification (Gal et al., 2015) and representation learning (Märtens et al., 2019) tasks.

## 3   Stochastic Active Sets

Our key objective is a computationally efficient estimate of the log-marginal likelihood in Eq. 2, as this is known to be a good measure of generalization performance (Rasmussen and Ghahramani, 2000; Germain et al., 2016). Another popular measure of generalization performance is *cross validation* (CV) (Geisser and Eddy, 1979; Vehtari and Lampinen, 2002), which is arguably mostly used outside the realm of Bayesian models. Recently, Fong and Holmes (2020) linked these two measures, effectively showing that the marginal likelihood is equivalent to the average over exhaustive leave-$R$-out CV scores. In particular, the average is w.r.t. the size of the hold-out set. More precisely, let

$$\mathcal{S}_{\mathrm{CV}}(\boldsymbol{x}|R) = \frac{1}{\mathcal{C}}\sum_{p=1}^{\mathcal{C}}\frac{1}{R}\sum_{n \in \mathcal{R}_p}\log p(\boldsymbol{x}_n|\boldsymbol{x}_{\mathcal{A}_p}, \boldsymbol{z}) = \frac{1}{R}\mathbb{E}_{\mathcal{A}_p}\left[\sum_{n \in \mathcal{R}_p}\log p(\boldsymbol{x}_n|\boldsymbol{x}_{\mathcal{A}_p}, \boldsymbol{z})\right], \tag{3}$$

denote the leave-$R$-out CV using log-predictive scoring functions $\log p(\boldsymbol{x}_n|\boldsymbol{x}_{\mathcal{A}_p}, \boldsymbol{z})$. Here $\mathcal{A}_p$ denotes the *active set* indices of the training data, such that $\mathcal{A}_p \subset \{1, 2, \dots, N\}$ and $\mathcal{R}_p = \{1, 2, \dots, N\}\backslash\mathcal{A}_p$ are the remaining hold-out samples. The subscript $p \in \mathcal{C}$ denotes the permutation, and we average over all $\mathcal{C} = \binom{N}{R}$ possible hold-out sets. We use use $R$ to indicate the size of the hold-out set $\mathcal{R}_p$ ($R \equiv |\mathcal{R}_p|$) and let $A \equiv |\mathcal{A}_p| = N - R$. If we average $\mathcal{S}_{\mathrm{CV}}(\boldsymbol{x}|R)$ over all possible sizes of the hold-out set, then Fong and Holmes (2020) has shown that we recover the exact log-marginal likelihood,

$$\log p(\boldsymbol{x}|\boldsymbol{z}) = \sum_{r=1}^{N}\mathcal{S}_{\mathrm{CV}}(\boldsymbol{x}|r) = \mathcal{S}_{\mathrm{CCV}}(\boldsymbol{x}|R) + \mathcal{S}_{\mathrm{PCV}}(\boldsymbol{x}|R). \tag{4}$$

Here, $\mathcal{S}_{\mathrm{PCV}}(\boldsymbol{x}|R) = \mathbb{E}_{\mathcal{A}}[\log p(\boldsymbol{x}_{\mathcal{A}}|\boldsymbol{z}_{\mathcal{A}})]$ and $\mathcal{S}_{\mathrm{CCV}}(\boldsymbol{x}|R) = \sum_{r=1}^{R}\mathcal{S}_{\mathrm{CV}}(\boldsymbol{x}|r)$ is the cumulative CV score, which reduces to a sum of expectations over the predictive factors.[3] Further details on $\mathcal{S}_{\mathrm{PCV}}$ and $\mathcal{S}_{\mathrm{CCV}}$ can be found in the Appendix. Fong and Holmes (2020) used Eq. 4 to argue in favor of using the marginal likelihood over cross-validation for model selection.

---

[3] We drop the permutation subscript $p$ in $\mathcal{A}$ and $\mathcal{R}$ to avoid cluttered notation.

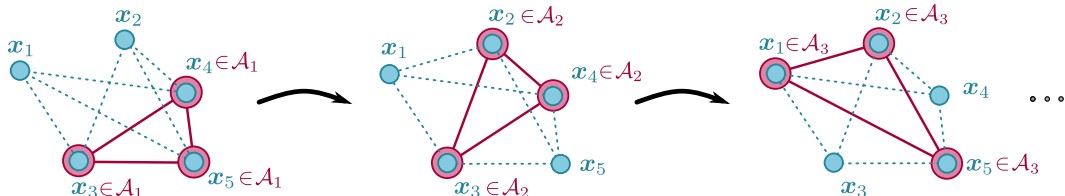

Figure 1: Schematic graphical model of the correlation structure given different permutations $p$ of the active set $\mathcal{A}_1, \mathcal{A}_2, \ldots, \mathcal{A}_\mathcal{C}$ for five observations $\{\boldsymbol{x}_1, \ldots, \boldsymbol{x}_5\}$. Thick red lines indicate that we build *full* covariance densities between observations included in $\boldsymbol{x}_\mathcal{A}$. Only-blue variables are considered conditionally independent among them w.r.t. to the red colored ones. Dashed lines indicate the conditional probability factors $p(\boldsymbol{x}_n|\boldsymbol{x}_\mathcal{A}, \boldsymbol{z})$ that we can *easily* compute.

**Stochastic approximation.** We take a slightly different view than Fong and Holmes (2020) and argue that Eq. 4 can be the grounds for an efficient stochastic gradient (Robbins and Monro, 1951) of the log-marginal likelihood suitable for training. In the context of GPs, we have that conditional probabilities for $\mathcal{S}_{\text{PCV}}(\boldsymbol{x}|R)$ and $\mathcal{S}_{\text{CCV}}(\boldsymbol{x}|R)$ in Eq. 4 are

$$p(\boldsymbol{x}_\mathcal{A}|\boldsymbol{z}_\mathcal{A}) = \mathcal{N}(\boldsymbol{x}_\mathcal{A}|0, \boldsymbol{K}_{\mathcal{A}\mathcal{A}} + \sigma_n^2\mathbb{I}), \qquad p(\boldsymbol{x}_n|\boldsymbol{x}_\mathcal{A}, \boldsymbol{z}) = \mathcal{N}(\boldsymbol{x}_n|\boldsymbol{m}_{n|\mathcal{A}}, \boldsymbol{c}_{n|\mathcal{A}}), \qquad (5)$$

where we used Eq. 2.22 from Rasmussen and Williams (2006) to obtain

$$\boldsymbol{m}_{n|\mathcal{A}} = \boldsymbol{K}_{n\mathcal{A}}(\boldsymbol{K}_{\mathcal{A}\mathcal{A}} + \sigma_n^2\mathbb{I})^{-1}\boldsymbol{x}_\mathcal{A},$$
$$\boldsymbol{c}_{n|\mathcal{A}} = \boldsymbol{K}_{nn} + \sigma_n^2\mathbb{I} - \boldsymbol{K}_{n\mathcal{A}}(\boldsymbol{K}_{\mathcal{A}\mathcal{A}} + \sigma_n^2\mathbb{I})^{-1}\boldsymbol{K}_{n\mathcal{A}}^\top,$$

and $\boldsymbol{K}_{\mathcal{A}\mathcal{A}} \in \mathbb{R}^{A \times A}$ has entries $k(\boldsymbol{z}_i, \boldsymbol{z}_j)$ with $\boldsymbol{z}_i, \boldsymbol{z}_j \in \boldsymbol{z}_\mathcal{A}$. The computational cost of $\mathcal{S}_{\text{PCV}}(\boldsymbol{x}|R)$ is $\mathcal{O}(A^3)$, while $\log p(\boldsymbol{x}_n|\boldsymbol{x}_\mathcal{A}, \boldsymbol{z})$ can reuse the matrix inversion $\boldsymbol{K}_{\mathcal{A}\mathcal{A}}^{-1}$ to only have an additional linear cost. Clearly, we can obtain an *unbiased* stochastic estimate of the log-marginal likelihood by *first* uniformly sampling $R$, *second* making a random split permutation into train and hold-out data $\mathcal{R}$ and *finally* by evaluating

$$\log p(\boldsymbol{x}|\boldsymbol{z}) \approx \sum_{n \in \mathcal{R}} \log p(\boldsymbol{x}_n|\boldsymbol{x}_\mathcal{A}, \boldsymbol{z}) + \log p(\boldsymbol{x}_\mathcal{A}|\boldsymbol{z}_\mathcal{A}), \qquad (6)$$

where we remark that the summation can be mini-batched. This approach is equivalent to the decomposition $\log p(\boldsymbol{x}|\boldsymbol{z}) = \log p(\boldsymbol{x}_\mathcal{R}|\boldsymbol{x}_\mathcal{A}, \boldsymbol{z}) + \log p(\boldsymbol{x}_\mathcal{A}|\boldsymbol{z}_\mathcal{A})$ and it also assumes conditional independence among observations $\boldsymbol{x}_n$ for $n \in \mathcal{R}$. This is similar to the standard *active set* approximation (Seeger et al., 2003) previously discussed, and we may think of $\mathcal{A}$ as a *stochastic active set (SAS)*.

However, the estimate of $\log p(\boldsymbol{x}|\boldsymbol{z})$ still has the same computational complexity $\mathcal{O}(N^3)$ as the usual deterministic approach, since we may sample $A = N$ in the worst case. Instead, we propose to make a stochastic approximation where we choose the size of the active set deterministically through a user-specified parameter, such that the computational cost reduces to $\mathcal{O}(A^3)$, as in sparse approximations based on inducing points. This does not need to be unbiased; empirically we find that most often the approximation behaves as a lower bound to the true marginal likelihood, and that, in all instances, it is a rather close approximation. We include a longer discussion on this point in the Appendix and the training methodology using SAS is in Alg. 1.

### 3.1 Extension for Bayesian GP decoders

We next seek to extend the previous SAS approach to the Bayesian GP-LVM, where we aim to obtain the posterior distribution $p(\boldsymbol{z}|\boldsymbol{x})$. From this perspective, we are interested in marginalising latent variables $\boldsymbol{z}$ to obtain the marginal likelihood $p(\boldsymbol{x})$ of the model.[4] However, this integration is not possible, as latent variables appear non-linearly in the kernel function (Titsias and Lawrence, 2010). Alternatively, we consider the variational inference scheme, where an auxiliary distribution $q(\boldsymbol{z})$ is

---

[4]Notice that the probabilistic objective function changes between standard and Bayesian versions of the GP-LVM. In the former case, we usually look for $p(\boldsymbol{x}|\boldsymbol{z})$ as the marginal likelihood w.r.t. the mapping $f$. This is the one usually considered in supervised GP tasks. In the latter, we refer to $p(\boldsymbol{x})$ as the marginal likelihood of the model, since $\boldsymbol{z}$ are also integrated out.

introduced into the formulation. Therefore, we are able to build the evidence lower bound (ELBO) of the model using Jensen's inequality as

$$\log p(\boldsymbol{x}) \geq \int q(\boldsymbol{z}) \log p(\boldsymbol{x}|\boldsymbol{z}) p(\boldsymbol{z}) \mathrm{d}\boldsymbol{z} = \mathbb{E}_{q(\boldsymbol{z})} \left[ \log p(\boldsymbol{x}|\boldsymbol{z}) \right] - \mathrm{KL} \left[ q(\boldsymbol{z}) || p(\boldsymbol{z}) \right], \quad (7)$$

which is equivalent to the one obtained by Titsias and Lawrence (2010, Eq. 8). At this point, the computational cost is $\mathcal{O}(N^3)$, since the ELBO requires evaluating $\log p(\boldsymbol{x}|\boldsymbol{z})$, where we invert $\boldsymbol{K}_{NN}$. The expectation in Eq. 7 can also benefit from a stochastic SAS approximation, just as with inducing points (Hensman et al., 2013). Thus, the lower bound can be approximated as

$$\mathcal{L}_{\text{ELBO}} \approx \sum_{n \in \mathcal{R}} \mathbb{E}_{q(\boldsymbol{z}_n)} \left[ \log p(\boldsymbol{x}_n | \boldsymbol{x}_{\mathcal{A}}, \boldsymbol{z}) \right] + \mathbb{E}_{q(\boldsymbol{z}_{\mathcal{A}})} \left[ \log p(\boldsymbol{x}_{\mathcal{A}} | \boldsymbol{z}_{\mathcal{A}}) \right] - \sum_{n=1}^{N} \mathrm{KL}[q(\boldsymbol{z}_n) || p(\boldsymbol{z}_n)], \quad (8)$$

where we consider *mean-field* VI to factorize the distribution $q(\boldsymbol{z})$. The Bayesian GP-LVM shares high-level similarities with other generative models that marginalize the latent variable according to a simple prior (Rezende and Mohamed, 2015). The proposed SAS approximation (8) scales similarly to mini-batched inducing point approximations, but we will later see that SAS behaves notably better in practice. Algorithmically, the approach is simple, and the summary code is provided in Alg. 2.

---

**Algorithm 1** SAS for GP decoders

1: **Input:** Observed data $\boldsymbol{x}$
2: **Parameters:** Initialize $\boldsymbol{\theta}, \boldsymbol{\phi}$      // $\boldsymbol{\theta}, \boldsymbol{z}$ if NA
3: **for** $e$ **in** epochs **do**
4:   **for** $b$ **in** batches **do**
5:     Sample $\boldsymbol{x}_{\text{batch}} \sim \boldsymbol{x}$
6:     $\boldsymbol{x}_{\mathcal{R}}, \boldsymbol{x}_{\mathcal{A}} \leftarrow \texttt{random\_split}(\boldsymbol{x}_{\text{batch}})$
7:     **if** amortized **then**
8:       Get $\{\boldsymbol{z}_{\mathcal{R}}, \boldsymbol{z}_{\mathcal{A}}\} \leftarrow g(\boldsymbol{x}_{\mathcal{R}}, \boldsymbol{x}_{\mathcal{A}} | \boldsymbol{\phi})$
9:     **end if**
10:    Compute $\boldsymbol{K}_{\mathcal{A}\mathcal{A}}^{-1}$      // via Cholesky
11:    Evaluate $\log p(\boldsymbol{x}_{\mathcal{A}} | \boldsymbol{z}_{\mathcal{A}})$
12:    Evaluate $\log p(\boldsymbol{x}_n | \boldsymbol{x}_{\mathcal{A}}, \boldsymbol{z}), \ \forall \boldsymbol{x}_n \in \boldsymbol{x}_{\mathcal{R}}$
13:    Evaluate Eq. 6
14:    **do** Adam$(\boldsymbol{\theta}, \boldsymbol{\phi})$ step for $\mathcal{L}$
15:   **end for**
16: **end for**

NA: Non-amortized.

**Algorithm 2** SAS for Bayesian GP decoders

1: **Input:** Observed data $\boldsymbol{x}$
2: **Parameters:** Initialize $\boldsymbol{\theta}, \boldsymbol{\phi}$  // $\boldsymbol{\theta}, \mu, \sigma$ if NA
3: **for** $e$ **in** epochs **do**
4:   **for** $b$ **in** batches **do**
5:     Sample $\boldsymbol{x}_{\text{batch}} \sim \boldsymbol{x}$
6:     $\boldsymbol{x}_{\mathcal{R}}, \boldsymbol{x}_{\mathcal{A}} \leftarrow \texttt{random\_split}(\boldsymbol{x}_{\text{batch}})$
7:     **if** amortized **then**
8:       Get $\mu_{\boldsymbol{z}} \leftarrow g_{\mu}(\boldsymbol{x}_{\mathcal{R}}, \boldsymbol{x}_{\mathcal{A}} | \boldsymbol{\phi})$
9:       Get $\sigma_{\boldsymbol{z}} \leftarrow g_{\sigma}(\boldsymbol{x}_{\mathcal{R}}, \boldsymbol{x}_{\mathcal{A}} | \boldsymbol{\phi})$
10:    **end if**
11:    Sample $\{\boldsymbol{z}_{\mathcal{R}}, \boldsymbol{z}_{\mathcal{A}}\} \sim q(\mu_{\boldsymbol{z}}, \sigma_{\boldsymbol{z}})$   // RT
12:    Compute $\boldsymbol{K}_{\mathcal{A}\mathcal{A}}^{-1}$      // via Cholesky
13:    Evaluate $\mathcal{L}$ in Eq. 8
14:    **do** Adam$(\boldsymbol{\theta}, \boldsymbol{\phi})$ step for $\mathcal{L}_{\text{ELBO}}$
15:   **end for**
16: **end for**

NA: Non-amortized, RT: Reparametrization trick.

---

### 3.2  The Role of Amortization

Early after the emergence of the seminal GP-LVM (Lawrence, 2005), the lengthy optimization required obtaining a result in which all latent representations $\boldsymbol{z}_n$ became noticeable. An additional consideration is that, while most approaches to non-linear low-dimensionality reduction focus on preserving similarities, the GPLVM does the opposite. This property was initially discussed by Lawrence and Quiñonero-Candela (2006), since in some sense the GP-LVM is *dissimilarity preserving*, such that different observations will generally be represented far away from each other. In practice, we are often more interested in embeddings that reflect the *true* distance between the observed objects in their representations, particularly those that are close together. This observation inspired *back constraints* for locality preservation (Lawrence and Quiñonero-Candela, 2006), which enforces latent variables $\boldsymbol{z}$ to be an explicit function of observations $\boldsymbol{z} = g(\boldsymbol{x}|\boldsymbol{\phi})$ parameterized by $\boldsymbol{\phi}$. This is similar to the stochastic *encoder* applied in variational autoencoders (Kingma and Welling, 2013; Rezende and Mohamed, 2015). This idea was also extended to the VI framework in GP-LVMs by Bui and Turner (2015) using a recognition model, e.g. $q(\boldsymbol{z}_n) = \mathcal{N}(\boldsymbol{z}_n | g_{\mu}(\boldsymbol{x}_n | \boldsymbol{\phi}), g_{\sigma}(\boldsymbol{x}_n | \boldsymbol{\phi}))$ and more recently, to accelerate hyperparameter learning in GPs with hierarchical attention networks (Liu et al., 2020).

In our context, we assume the mappings to be neural networks (NNs) like Bui and Turner (2015), referring to them as *amortization* networks. We find such networks accelerate learning very nicely when used in conjunction with SAS. It is also worth noting that amortization has been empirically shown to improve generalization performance (Shu et al., 2018).

# 4    Related Work

Marginal likelihood approximations were used in GPs (Smola and Bartlett, 2001; Csató and Opper, 2002) before the apparition of pseudo-inputs (Snelson and Ghahramani, 2006) and the associated variational inference framework (Titsias, 2009). In the former case, stochastic approximations to the ELBO were first presented by Hensman et al. (2013, 2015), in line with the Bayesian counterpart of SAS. In terms of active sets, Seeger et al. (2003) empirically observed that the approximation was generally stable for optimisation, even if the size of $\mathcal{A}$ was a small fraction of the training size only. However, they also observed that random selection of active sets led to non-smooth fluctuations, making it difficult to converge through exact gradient ascent. Particularly, the issue with re-selecting of $\mathcal{A}$ motivated Snelson and Ghahramani (2006) as a *smoother* optimization alternative, and we find that SAS also circumvents this problem via stochastic estimates as shown in Sec. 5.

The connection between cross-validation and GPs was already described in Rasmussen and Williams (2006) as an alternative for model selection. However, the equivalence between *exhaustive* CV and the log-marginal likelihood provided in Fong and Holmes (2020) provides a novel perspective that we exploit. Additionally, the notion of *back constraints* has recently been considered in Lalchand et al. (2022) for GP-LVMs with inducing points, where a doubly stochastic formulation is used. More recently, amortization networks have been used to drastically reduce the number of inducing points in supervised GPs.

# 5    Experiments

In this section, we evaluate the performance of the SAS approach for stochastic learning of GP decoders, both the deterministic GP-LVM (Lawrence, 2004) and its Bayesian counterpart (Titsias and Lawrence, 2010). For this purpose, we consider three different datasets: MNIST (LeCun et al., 1998), FASHION MNIST (Xiao et al., 2017) and CIFAR-10 (Krizhevsky, 2009). For a *fair* comparison of our model with baseline methods, we use the same amortization across models, namely a neural network (three linear layers ReLU activation functions)[5] for all models in all experiments and all GPs use an quadratic exponentiated kernel. In all experiments, the learning rates are set in the range $[10^{-4}, 10^{-2}]$, the maximum number of epochs considered is $300$ and we use the ADAM optimizer (Kingma and Ba, 2015). For SAS experiments, we only consider batch sizes greater than the active set size, as this is a requirement for SAS.

Performance metrics of the SAS-GP decoders are given in terms of the negative log-predictive density (NLPD), root mean-square error (RMSE) and mean absolute error (MAE). In all cases, we optimize w.r.t. an approximation to the log-marginal likelihood $\log p(\boldsymbol{x}|\boldsymbol{z})$ in the deterministic scenario or w.r.t. a lower bound on the $\log p(\boldsymbol{x})$ *of the model* in the Bayesian cases. We also provide PYTORCH code that allows for reproducing all experiments and models.[6] We monitor the run-time of convergence as we suspect that rotating active sets (see Fig. 1) across the dataset is a fast way to capture the correlation structure of the regression problem.

## 5.1    Representation performance

First, we analysed our SAS approach and the approximate optimization procedure on an *unsupervised* version of MNIST, FMNIST and CIFAR-10, where we took the full training corpus for learning two-dimensional latent representations of images. The approximation curves are included in Fig. 2, where we observe convergence in less that 2h runtime in CPU for most cases. For all experiments, we can observe that for larger active set sizes $A$, the SAS approaches take longer time to complete the 300 epochs, as the computational cost of inversion is higher.

**Evaluation with SOTA methods.**    We also tested the performance of representation learning in the state-of-the-art methods with and without GPs for unsupervised scenarios. Results are shown in Fig. 4. We include a short description of the models considered:

- SAS-GP DECODER — It uses stochastic active sets to approximate the log-marginal likelihood $\log p(\boldsymbol{x}|\boldsymbol{z})$. The training methodology is described in Alg. 1.

---

[5]Please see the supplementary material for more details.

[6]The code is publicly available in the repository: `https://github.com/pmorenoz/SASGP/` including baselines.

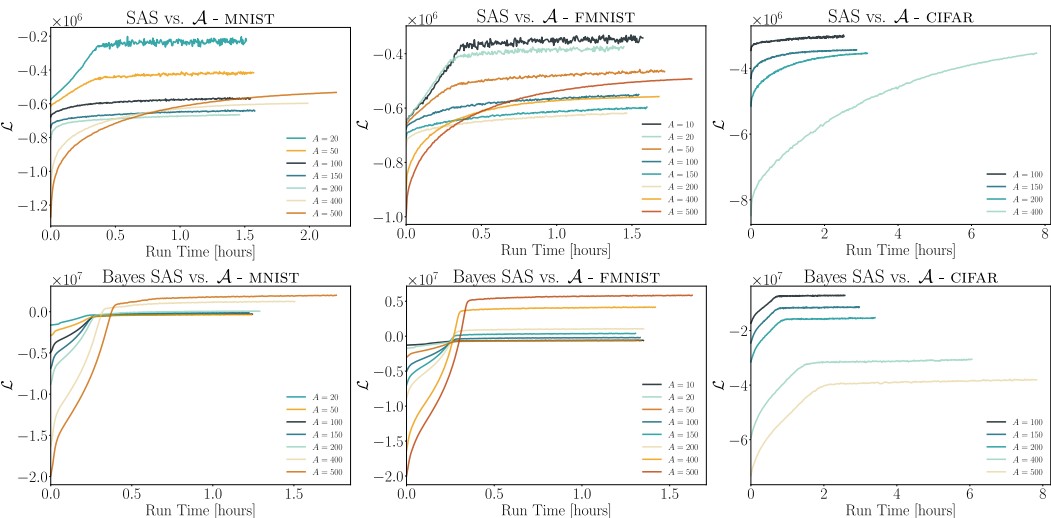

Figure 2: Approximate log-marginal likelihood (**upper row**) for SAS and ELBO curves (**lower row**) for Bayesian SAS. We fix the batch size in Alg. 1 to be $B = 1024$ and study the convergence for different *active set* sizes $A$. All values in the curves displayed are *per-epoch*.

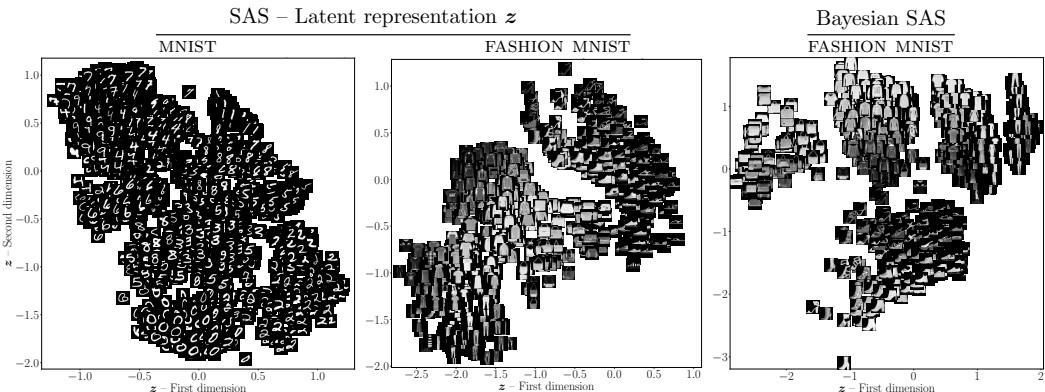

Figure 3: Latent representation in a 2-dimensional space $\mathcal{Z}$ for the 10 MNIST and FMNIST classes learnt with SAS and Bayesian SAS. Notice that the likelihood model of the GP decoder is controlled by a *vanilla* RBF kernel. The examples have been obtained using an *active set* size $A = 800$.

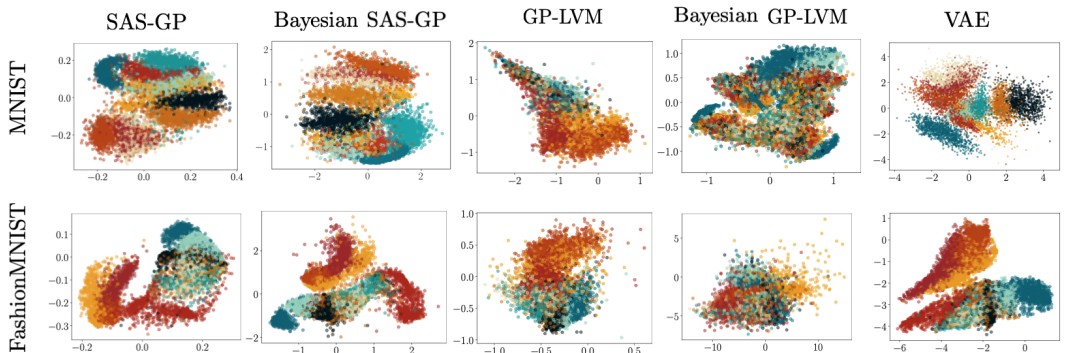

Figure 4: Illustration of latent space mappings $z_n \in \mathcal{Z}$ on test data for different SAS models and baselines (**rows**) and different datasets (**columns**). The models have been trained on *full* MNIST and FMNIST.

- BAYESIAN SAS-GP DECODER — It uses stochastic active sets to approximate the ELBO in Eq. (8) on $\log p(\boldsymbol{x})$. See Alg. 2.
- BAYESIAN GP-LVM — It is based on the model in Titsias and Lawrence (2010). Means and variance parameters are generated by an amortization NN as in Bui and Turner (2015). The model is trained using stochastic variational inference (Hensman et al., 2013).
- GP-LVM — We used the model proposed by Lawrence (2005) enhanced with an NN encoding to the latent space. The model is trained using maximum likelihood (ML).
- VARIATIONAL AUTOENCODER (VAE) — (Kingma and Welling, 2013) The NN encoder has the same architecture as the amortization function used in the SAS-GP models.

For the SAS-GP decoder and GP-LVM, a neural network encodes the latent variables. Likewise, for the Bayesian SAS-GP decoder and the Bayesian GP-LVM, two neural networks each encode the latent means and latent variances. We refer to this encoding as *amortization*. All models use a Gaussian likelihood[7].

Table 1: Comparative metrics for SAS and Bayesian SAS on MNIST, FMNIST and CIFAR-10.

| MODEL | SAS | | | BAYESIAN SAS | | |
|---|---|---|---|---|---|---|
| ACTIVE SET SIZE | $A = 100$ | $A = 200$ | $A = 400$ | $A = 100$ | $A = 200$ | $A = 400$ |
| MNIST / RMSE ↓ | $2.55 \pm 0.98$ | $2.47 \pm 0.98$ | $2.41 \pm 0.93$ | $2.16 \pm 0.02$ | $2.08 \pm 0.02$ | $1.99 \pm 0.02$ |
| MNIST / MAE ↓ | $1.61 \pm 0.97$ | $1.55 \pm 0.99$ | $1.51 \pm 0.96$ | $1.11 \pm 0.02$ | $1.04 \pm 0.02$ | $0.96 \pm 0.01$ |
| MNIST / NLPD ↓ | $2.99 \pm 1.41$ | $2.92 \pm 1.38$ | $2.84 \pm 1.31$ | $2.33 \pm 0.03$ | $2.26 \pm 0.02$ | $2.17 \pm 0.02$ |
| FMNIST / RMSE ↓ | $2.37 \pm 0.95$ | $2.31 \pm 0.94$ | $2.25 \pm 0.90$ | $1.99 \pm 0.17$ | $1.88 \pm 0.20$ | $1.85 \pm 0.13$ |
| FMNIST / MAE ↓ | $1.48 \pm 0.91$ | $1.42 \pm 0.91$ | $1.39 \pm 0.89$ | $1.11 \pm 0.02$ | $1.02 \pm 0.03$ | $0.98 \pm 0.02$ |
| FMNIST / NLPD ↓ | $2.76 \pm 1.33$ | $2.71 \pm 1.31$ | $2.65 \pm 1.23$ | $2.16 \pm 0.18$ | $2.07 \pm 0.19$ | $2.04 \pm 0.12$ |
| CIFAR10 / RMSE ↓ | $2.66 \pm 1.08$ | $2.55 \pm 1.06$ | $2.55 \pm 1.03$ | $2.74 \pm 1.07$ | $2.64 \pm 1.08$ | $2.57 \pm 1.02$ |
| CIFAR10 / MAE ↓ | $1.77 \pm 1.06$ | $1.69 \pm 1.06$ | $1.69 \pm 1.02$ | $1.84 \pm 1.03$ | $1.76 \pm 1.05$ | $1.71 \pm 1.03$ |
| CIFAR10 / NLPD ↓ | $3.20 \pm 1.55$ | $3.07 \pm 1.44$ | $3.32 \pm 1.89$ | $3.24 \pm 1.53$ | $3.14 \pm 1.53$ | $3.06 \pm 1.45$ |

All metrics are $(\times 10^{-1})$.

## 5.2 Evaluation metrics

In this section, we are interested in the evaluation of the GP decoder with the *standard* error metrics used for GPs. In Tab. 1 we provide RMSE, MAE and NLPD for the three datasets considered in this experiment. Interestingly, the performance usually improves with larger active set sizes $A$, as the SAS model captures captures the underlying correlation of datapoints better and this enhances the approximation. This happens for both deterministic and Bayesian cases. Additionally, we observe that the NLPD metric is generally better for smaller active set sizes $A$. This can also be noticed in Fig. 2, where loss curves have a better convergence for the lowest $A$.

**Classification accuracy.** We are interested in evaluating the structure of the representation. For this purpose, we trained a (one) *nearest neighbour* classifier on the encoded, two-dimensional latent variables and we tested the accuracy using encoded test data. Tab. 2 shows the mean and standard deviations of test the accuracy.

Table 2: Classification accuracy (↑) on 2-dim. latent space $\mathcal{Z}$.

| MODEL | MNIST | FMNIST |
|---|---|---|
| BAYESIAN SAS-GP DEC. (**ours**) | $0.63 \pm 0.022$ | $0.63 \pm 0.020$ |
| BAYESIAN GP-LVM | $0.18 \pm 0.033$ | $0.24 \pm 0.043$ |
| VAE | $0.54 \pm 0.026$ | $0.58 \pm 0.008$ |

**Runtime and convergence of SAS approximation.** The loss curves obtained for the (sparse) Bayesian GP-LVM model and the Bayesian SAS-GP decoder are shown in Fig. 5. We observe that the two methods scale similarly, but that SAS is faster by a notable constant within stochastic optimization. Additionally, we observe the performance of the SAS approximation to closely fit the exact log-marginal likelihood $\log p(\boldsymbol{x}|\boldsymbol{z})$. For the plot in Fig. 5, we computed the exact probability for a subset of MNIST $N=40,000$ samples.

---

[7]Pytorch (Paszke et al., 2019) and Pyro (Bingham et al., 2019) are available on public repositories.

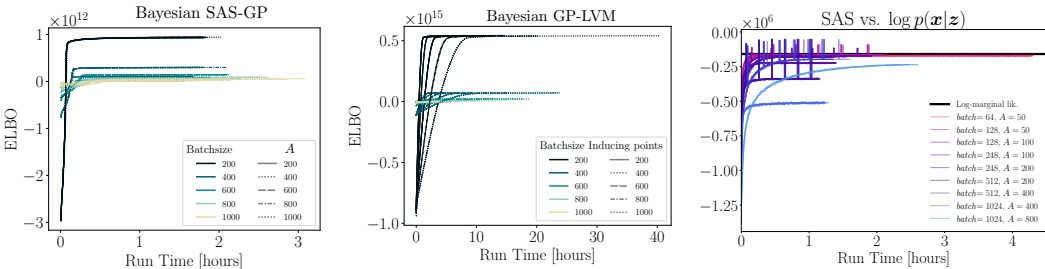

Figure 5: (**Left-center**) ELBO curves for Bayesian SAS-GP and Bayesian GPLVM for different batch-sizes and active set points. (**Right**) SAS loss function compared with the exact log-marginal likelihood computed. In all curves, $N{=}40,000$ samples of MNIST were considered and five different initializations per *batch* and $A$ setup.

## 6    Conclusion

State-of-the-art representation learning is generally based on neural networks, as this allows for scaling to large datasets. However, often we want reliable uncertainty estimates from the model and we can achieve these with Gaussian process decoders if we can scale them sufficiently. We have reviewed the main difficulties to obtain decent performances with GP-LVM approaches when applied to large-scale learning, even with inducing variables. Revisiting active set approximations, we considered a stochastic viewpoint to approximate the marginal likelihood while simultaneously keeping the model marginalized. We formulated our *stochastic active sets (SAS)* approach for both deterministic and Bayesian versions of GP decoders. We found that our approach works well with amortization, such that a neural network encoder approximately inverts the GP decoder. While amortization also helps when using inducing points, we found the combination with SAS to be particularly efficient and robust.

Empirically, we illustrated the advantage of our method first on image-based observations, where our approach learns better representations using fewer computational resources compared to inducing point methods. We further demonstrated that our approach easily scales to nearly $10^6$ observations. In this experiment, we found that the learnt representations are qualitatively on par with those attained by a comparable autoencoder. This is an important finding as, beyond small datasets, GP decoders generally recovers less useful representations compared with models based on neural networks. From this result, we speculate that improvements in *training* might be enough to get state-of-the-art representations with GP decoders.

**Additional benefits.**    Besides the empirical benefits demonstrated by SAS in the previous section, we have also observed other practical benefits worth reporting. *First*, we have observed that SAS easily runs in 32-bit numerical precision, unlike inducing point methods that generally require 64-bits of precision (when reporting running times we consistently used 64 bits). Similarly, the *jitter* usually added to the Cholesky factorization is of less importance in SAS. *Second*, we note that our implementation is surprisingly free of additional *tricks* and no numerical heuristics were needed to realize a reliable implementation.

**Limitations and future work.**    Stochastic active sets rely on the Gaussian likelihood, and this is perhaps the strongest limitation. This works well for continuous data, but many data sources are inherently discrete and this requires a suitable likelihood, e.g. the discretized mixture of logistics (Salimans et al., 2017). Having more powerful likelihoods would surely improve the GP decoders, but this requires realization of further developments using SAS.

Future work will focus on applying the SAS approach in the supervised setting as well, and building SAS-like methods for discrete data. Other possible directions include extending the decoder with deep kernels (Wilson et al., 2016) to capture more features in the data and applying convolutional GPs (Van der Wilk et al., 2017) which are more suited to high-dimensional images.

## Acknowledgements

The authors want to thank Simon Bartels for the fruitful discussions during the early stages of our investigation. This work was supported by research grants (15334, 42062) from VILLUM FONDEN. This project has also received funding from the European Research Council (ERC) under the European Union's Horizon 2020 research and innovation programme (grant agreement 757360). This work was funded in part by the Novo Nordisk Foundation through the Center for Basic Machine Learning Research in Life Science (NNF20OC0062606). This work was further supported by the Pioneer Centre for AI, DNRF grant number P1.

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
