# OpenReview forum: "Revisiting Active Sets for Gaussian Process Decoders"
_NeurIPS.cc/2022/Conference — NeurIPS 2022 Accept_

### Official Review · Reviewer_38P8 · 2022-06-30

**Rating:** 7
**Confidence:** 3
**Soundness:** 3 good
**Presentation:** 4 excellent
**Contribution:** 3 good

**Summary:**

The authors investigate a novel stochastic gradient ascent method for fitting Gaussian Process Latent Variable Models. In practice, if I understood correctly, the method samples a "minibatch" from the full data to build a GP, and measures posterior likelihood on the remaining hold out data. This is justified by recent mathematical results that show the full log marginal likelihood ($O(n^3)$ cost) is equivalent to the marginal likelihood over subsets ($O(b^3)$ with batchsize $b$) of all subset sizes and ( $O(n-b)$, linear cost) summations over terms excluded from the subset. This alternative marginal likelihood evaluation is a a sum and an expectation and is amenable to stochastic Monte-Carlo approximation, and therefore unbiased stochastic gradient ascent.

The authors apply this trick to both standard and Bayesian GP-LVMs implemented both with and without amortization over a range of datasets and demonstrate the benefits.

**Questions:**

- in the Standard GP-LVM case, can the authors confirm that the only practical novelty (over naïve subsampling data) is the random batch size and additional term in Equation (6) and (8)? (If so, this can be portrayed as a strength)
- Can the authors include such a naive subsampling data baseline forming an ablation study?

**Limitations:**

The authors clearly discuss the Gaussian likelihood assumption. No more unaccounted limitations come to my mind.

**Strengths And Weaknesses:**

**Strengths**
- I felt the paper was very well written, providing a clear story of GP-LVMs and their milestones over the years and with clear interpretation
- the result of Fong and Holmes 2020 appears to show the marginal likelihood can be (stochastically) approximated using just partitions of data, which is very impactful for GPs that require cubic cost effectively reducing cost to smaller cubes and linear costs.
- I felt the application to GP-LVMs and Bayesian prior GP-LVMs with and without amortization covered a wide extensive range of empirical use cases and provides value to the community.

**Weaknesses**
- I feel that evaluating marginal likelihood on a stochastically subsampled data to reduce the cost of GP training is a fairly common trick, but obviously biased. Admittedly I cannot seem to find papers using it exactly, a google search for "Gaussian Process minibatch" returned some results [1, 2].
- Hence the only _practical_ novelty (over naive subsampling) in this work is
    - (1) the first term of Equations 6 and Equation 8 and
    - (2) the batch size is random (but it doesn't need to be? See nitpick)
- The rest of the paper seems to be technical background on GP-LVMS and Fong and Holmes (2020).
- Can the authors include a naive random subsampling baseline? Simply SAS methods without the First terms of equation 6? This would serve as an ablation to show the impact of the theoretical result.

**Nit Picks**
- Equation 4: $R$ is said to be averaged over $R$ yet it is a constant on the right term but not in the centre or left terms, this is explained in the Appendix "Additionally, Eq. (3) holds for every size of the hold-out data R ∈ [1, 2, · · · , N]" and would make Eqn 4 easier less confusing to me.
- Equation 3: for me, seeing the bold $x_n$ automatically made me think it was a set of correlated points requiring whose probability would have led to $n^3$ matrix inversion.
- Appendix Equation 4: I was very curious to know the intermediate steps to deriving $S_{PCV}(x, R)$ and I couldn't find the answer during a (very) quick skim through Fong and Holmes (2020).


[1] https://arxiv.org/abs/2111.10461

[2] https://nipunbatra.github.io/blog/ml/2021/09/03/param-learning-sgd.html

---

> ### Author Response · Authors · 2022-08-02
> **Response to Reviewer 38P8 -- Part (2/2)**
>
> **Point 5 / Nit Picks**
> > *Equation 4: $R$ is said to be averaged over $R$ yet it is a constant on the right term but not in the centre or left terms, this is explained in the Appendix "Additionally, Eq. (3) holds for every size of the hold-out data $R \in [1, 2, · · · , N]$" and would make Eqn 4 easier less confusing to me.*
>
> If we understand your comment correctly, then you suggest that we add the information that equation (4) holds for all $R$. We are planning a major update of Section 3 based on the feedback and we will surely take this into account.
>
> **Point 6 / Nit Picks**
> > *Equation 3: for me, seeing the bold $\boldsymbol{x}_n$ automatically made me think it was a set of correlated points requiring whose probability would have led to  matrix inversion.*
>
> We intended to use the $\boldsymbol{x}_n$ as a vector containing just one data point. We will make sure to clarify this.
>
> **Point 7 / Nit Picks**
> > *Appendix Equation 4: I was very curious to know the intermediate steps to deriving  and I couldn't find the answer during a (very) quick skim through Fong and Holmes (2020).*
>
> We share this curiosity! Extra details on the derivation can be found in the supplementary material of Fong and Holmes (2020)$$S_{PCV} = \frac{1}{\binom{n}{P}} \sum_{t=1}^{\binom{n}{P}} \log p_M (\boldsymbol{y}_{1:n-P}^{(t)}).
> $$
> They show by induction and they continue by arguing that setting $P=0$ recovers Proposition 2.
>
>
> **Point 8 / Questions**
> >*in the Standard GP-LVM case, can the authors confirm that the only practical novelty (over naïve subsampling data) is the random batch size and additional term in Equation (6) and (8)? (If so, this can be portrayed as a strength)*
>
> Yes. (Just to be clear, we also use an amortization network for the latent variables).
>
>
> **Point 9 / Questions**
> >*Can the authors include such a naive subsampling data baseline forming an ablation study?*
>
> We have done extra experiments for this question, which are included in the Appendix now. In particular, we invite the reviewer to check the results of the ablation study in Figures 2 and 3 of the Appendix. We appreciate the curiosity, and from the early results, we can confirm that the performance of both terms individually is lower than in our framework. Obviously, the structure learnt by the full-covariance term seems to be better for both MNIST and FMNIST.

---

> > ### Comment · Reviewer_38P8 · 2022-08-10
> > **Thank you for your efforts**
> >
> > I apologize for missing the amortization, I believe amortization in GP models has been studied in multiple works for predicting a latent embedding, it seems the use case is slightly different.
> >
> > - the latent embeddings in this work is used as input to a decoder GP
> > - in [1] the latent embeddings have time stamps and embeddings are grouped each making a dataset for GP regression in latent space
> >
> > I am rather surprised that GP-LVM case has not been amortized before? The other reviewers have not raised this as a concern.
> >
> > I am happy all my concerns have been addressed.
> >
> > [1] https://proceedings.mlr.press/v130/jazbec21a.html

---

> > > ### Author Response · Authors · 2022-08-10
> > > **Final comment for Reviewer 38P8**
> > >
> > > We thank the reviewer for the acknowledgement on our response and for the positive consideration of our work. We are also glad to hear that all concerns are now clarified.
> > >
> > > As a final comment, we wanted to remark that the connection between the GP-LVM and amortization has been considered before in the literature, but usually under the name of *"back constraints"*. This name is due to the work of *Lawrence and Quiñonero-Candela, (2006)*, which focused on the locality preservation of the GP-LVM. The idea was later explored by *Bui and Turner (2015)*, in their workshop paper. We carefully reviewed this and all references and details can be found in the *Section 3.2* of the main paper.
> > >
> > > - *N. D. Lawrence and J. Quiñonero-Candela. Local distance preservation in the GP-LVM through back constraints, ICML, 2006*.
> > > - *T. D. Bui and R. E. Turner. Stochastic variational inference for Gaussian process latent variable models using  back constraints. In Black Box Learning and Inference Workshop @ NIPS, 2015.*

---

> ### Author Response · Authors · 2022-08-02
> **Response to Reviewer 38P8 -- Part (1/2)**
>
> We thank the reviewer for the positive consideration and the useful feedback on our work. We also appreciate the time you clearly took to review our paper. In the following, we addres the points raised under **Weaknesses**, **Nit Picks** and **Questions** individually.
>
> **Point 1 / Weaknesses**
> > *I feel that evaluating marginal likelihood on a stochastically subsampled data to reduce the cost of GP training is a fairly common trick, but obviously biased. Admittedly I cannot seem to find papers using it exactly, a google search for "Gaussian Process minibatch" returned some results [1, 2].*
>
> J. Hensman, N. Fusi and N. Lawrence developed stochastic variational inference in 2013 their paper *Gaussian Processes for Big Data*. However, this work only fits to the Bayesian GPLVM as it relies on variational inference and inducing points. To our knowledge, no previous connections has yet developed between the seminal sparse approximations based on active sets and stochastic optimization.
>
>
> Thanks for the additional references, that we have checked out. In particular, the blog post does not pick the points in the gradient randomly which surely would introduce a bias. We will consider the additional reference for an update of the related work.
>
>
> **Point 2 / Weaknesses**
> > *Hence the only practical novelty (over naive subsampling) in this work is
> (1) the first term of Equations 6 and Equation 8 and
> (2) the batch size is random (but it doesn't need to be? See nitpick)*
>
> For us is difficult to summarise the novelties in just two terms from two equations. But yes, the approximation to the log-marginal likelihood based on the factorisation on conditionals is one of the practical novelties, as shown in Eq. 6 and Eq. 8 for the Bayesian case. On the other hand, the presence of the amortization network together with the stochastic optimization schemes is also another point of practical novelty for us. The theoretical justification brought from the link with Fong & Holmes (2020) is also a novelty for GP-LVMs, in our opinion. Also the experimental results are demonstrating a good performance, which could be of interest for the community. Particularly those ones who need scalable models for probabilistic representation learning.
>
> **Point 3 / Weaknesses**
> > *The rest of the paper seems to be technical background on GP-LVMS and Fong and Holmes (2020).*
>
> We theoretically connected our work with Fong & Holmes (2020), which focus on a very different perspective to us. They indeed connect leave-$R$-out cross validation (CV) with log-marginal likelihood estimation, but without consider an especific model or stochastic methods for approximate optimization.
>
> **Point 4 / Weaknesses**
> > *Can the authors include a naive random subsampling baseline? Simply SAS methods without the First terms of equation 6? This would serve as an ablation to show the impact of the theoretical result.*
>
> This is a great suggestion. We have done extra experiments on this ablation study for Eq. 6. More details are included in the following lines, where the reviewer explicitly indicated that would be great to include this experiment.

---

### Official Review · Reviewer_YAoZ · 2022-07-11

**Rating:** 5
**Confidence:** 4
**Soundness:** 3 good
**Presentation:** 2 fair
**Contribution:** 3 good

**Summary:**

This paper presents a new approach to train Gaussian process latent variable models (GP-LVMs). The idea relies on the (previously shown) connection between cross-validation (CV) and marginal likelihood (ML). More specifically, the authors interpret the leave-R-out CV objective from an optimization perspective, which allows for stochastic and scalable optimization of the ML objective of vanilla and Bayesian GP-LVM formulations. The experiments demonstrate that the framework leads to more compact latent representations compared to previous formulations. On MNIST, the deterministic variant of the framework is shown to match the exact ML with a certain hyperparameter choice (batch and active set size).

**Questions:**

Presentation/clarity/notation:
- What is $K_{NN}$ in (1)?
- Shouldn't $S_{CV}(x|R)$ in (3) be also conditioned on $z$?
- How about including the formal definitions of $S_{PCV}$ and $S_{CCV}$ in the main paper instead of supplements?
- $r$ appears on the middle term in (4) but only $R$ appears on the right hand side.
- The first equation in line 114 involves $R$ on the left but not on the right. If I understood correctly, $|\mathcal{A}|=N-R$ (hence $R$ implicitly appears on the right. In that case, I would suggest using modifying the notation a bit to reflect this.
- Are $\mathcal{A}$ and $\mathcal{A}_p$ used interchangeably?

Experiments:
- The loss curves in Figure 2 oscillate a lot.
- The MNIST latent space with Bayesian SAS is missing.
- Only $2D$ latent spaces are considered, which significantly restricts the generalizability of the findings.
- NLPD consistently increases with active set size (for which I would use the shorthand $|\mathcal{A}|$ instead of $A$). Is this just the opposite of what we would expect?

Misc:
- Although the proposed framework seems suitable for GP models in general, the authors particularly focus on GP-LVMs. I wonder why authors restrict themselves to GP-LVM. It would also be nice to see how the framework performs on a standard GP regression task with simple noisy inputs and outputs.

**Limitations:**

Everything sounds good.

**Strengths And Weaknesses:**

Overall, I liked the paper as it nicely brings theoretical and practical aspects together. My major comments concern the derivation/notation and experiments, which strangely both escalate and diminish my score.

Strengths:
- The framework has solid theoretical foundations. I find the connection with CV important as it distinguishes the proposed framework from heuristics-based optimization objectives.
- The results are impressive. The improvement over existing deterministic and Bayesian GP-LVMs is very clear. Looking at the latent nearest neighbor classification experiment, one can even argue that the learned latent space is more structured than that of VAE.

Weaknesses:
- The presentation can be improved. I list several points in the subsequent "Question" section. Most importantly, I do not understand
  - how to obtain (5) from (4), and
  - how $S_{PCV}$ in (5) has $x_A$ on the left hand side of the conditioning while (3) has $x_A$ on the right.
- The claims in lines 29-34 are not well-supported. It would be much better to list the problems of training with inducing points (with references) instead of quotations.
- The experiments are slightly worrying (please see the next "Question" section again)

I would be glad to increase my score if my concerns are addressed (even verbally).

**UPDATE:** After the author response, I increase my score to 5.

---

> ### Author Response · Authors · 2022-08-02
> **Response to Reviewer YAoZ -- (Part 3/3)**
>
> **Point 11 / Experiments**
> > *Only 2D latent spaces are considered, which significantly restricts the generalizability of the findings.*
>
> Thanks for pointing this out. Similarly as we replied to Reviewer "Qibs", we have included two extra experiments with three and four latent dimensions.  Figures can be accessed in the Appendix, which we also updated. We want to remark that our implementation is not restricted by the dimensionality of the latent space, and indeed we had it as a parameter when training the model.
>
> ```
> parser.add_argument('--latent_dim', default=2, type=int, help='dimensionality of the latent space')
> ```
>
> **Point 12 / Experiments**
> > *NLPD consistently increases with active set size (for which I would use the shorthand $|\mathcal{A}|$ instead of $\mathcal{A}$). Is this just the opposite of what we would expect?*
>
> Good catch. We had to understand this as well during the rebuttal, and we found out a little typo in the NLPD equation that was making this to do the opposite as expected (RMSE and MAE were performing well). Thus, we have re-computed all experiments and metrics and updated the entire Table 1. Now metrics are consistently decreasing with the active set size, as expected, as similarly as happens with the size of inducing points in sparse GPs.
>
> **Point 13 / Misc**
> > *Although the proposed framework seems suitable for GP models in general, the authors particularly focus on GP-LVMs. I wonder why authors restrict themselves to GP-LVM. It would also be nice to see how the framework performs on a standard GP regression task with simple noisy inputs and outputs.*
>
> This is a great point, and we appreciate your curiosity very much. Initially, we were interested in probabilistic models for representation learning. This made us to focus on GP-LVMs, from the beginning. Moreover, our initial aim was to scale them up in a modern way (i.e. using stochastic optimization), and we finally built the link with Fong & Holmes (2020) which led us for the rest of contributions. This somehow explains why we only focused on the unsupervised case of GPs for this particular work.
>
> Additionally, we are currently investigating the performance of SAS for standard GP regression tasks, but so far, the results indicate that it make the optimization harder if we are not considering the amortization network as in the GP-LVM. We suspect that recent advances on those lines (Jafrasteh, ICML'22), would combine very well with SAS.
>
> - *B. Jafrasteh et al. Input Dependent Sparse Gaussian Processes, ICML, 2022.*

---

> > ### Comment · Reviewer_YAoZ · 2022-08-06
> > **.**
> >
> > Thanks for your detailed response. I must give your credit in that you gave satisfactory answers to most of the points I raised. I increase my score to "borderline accept". To give you an idea of your reply and my current take:
> > - Your elaborate derivation of (5) seems accurate, thanks for the clarification! This was one of my main concerns, which is now fully addressed.
> > - I believe an extensive re-write of Section 3 would be great. My main suggestion would be to include definitions and even abbreviations (e.g. preparatory CV) in the main text, and also include intuitive (verbal) definitions as well. These suggestions are, of course, tied with my comprehension and may not generalize to a common understanding. This is why I also suggest getting feedback on this from other reviewers & AC. These being said, it would have been much better to see the new version of the write-up to completely eliminate my "clarity" concern (which would probably be extremely difficult for you given the short rebuttal time). I promise to discuss with other reviewers this point before finalizing my score.
> > - Figure 2 is now much easier to read. Yet, I suggest using an alternative color palette and sorting the legend entries by magnitude, not alphabetically.
> > - I still think Figure 3 is somehow funny as we see two alternative models. Yet one is visualized on 2 datasets and the other only on 1 dataset.
> > - Thanks for the additional experiments. Did I get this right: Table 1 says bigger A's are more favorable while Figures 1 and 2 in the appendix favor smaller A's. If so, isn't this contradictory?
> > - Interesting that SAS makes GP regression training more challenging. I still think the framework applied only on GP-LVM slightly restricts its impact, which I will again discuss with other reviewers.

---

> > > ### Author Response · Authors · 2022-08-07
> > > **Extra comments for Reviewer YAoZ**
> > >
> > > We thank again the reviewer for the positive consideration, the time spent on reading our detailed response, and also for the update of the score. We particularly appreciate the help for polishing and improving the clarity of the paper, which is very important for us. We add some extra comments to the points mentioned:
> > >
> > > **Extra Point 1**
> > >
> > > > *I believe an extensive re-write of Section 3 would be great. (...)*
> > >
> > > - On the structure of Section 3, we were not 100% sure how interesting the link with Fong & Holmes (2020) could be for the potential reader of our work, when we wrote the manuscript before the submission. That's one of the main reasons why this section is more focused on the standard Gaussian Process view and notation than on the CV formulation. However, we are glad (and positively surprised) that the reviewers liked this connection a lot, considering it a main point of strength. So we are open to update it as the reviewer is recommending if the paper goes on in the process.
> > >
> > > **Extra Point 2**
> > >
> > > > *Figure 2 is now much easier to read. Yet, I suggest (...)*
> > >
> > > - The reviewer is absolutely right. We will update Figure 2 with a new color palette and legend as soon as possible. It will be ready before the end of the discussion period.
> > >
> > > **Extra Point 3**
> > >
> > > > *Table 1 says bigger A’s are more favorable while (...)*
> > >
> > > Right, the effect of increasing the active set $A$ *should* be similar as the effect of increasing $M$ (number of inducing points) in variational sparse GPs, as we are building better approximations to the *log-marginal likelihood*. This is usually visualised through the reduction of the RMSE in the SOTA, and in our case, also in the rest of metrics. If we look to Fig. 1 and 2 in the Appendix, is true what the reviewer observe, but we do not think that these curves are always comparable, as for every size $A$, we have a different *biased* estimator of the log-marginal likelihood.
> > >
> > > Additionally, these extra figures were representative of the convergence of the experiments with latent spaces of 3 and 4 dimensions, and we only plotted one trial. It could be also the case that the initialisation affects the optimization. We will complete these extra experiments also with error metrics if the paper goes on.
> > >
> > > **Extra Point 4**
> > >
> > > > Interesting that SAS makes GP regression training more challenging. I still think (...).
> > >
> > > Thanks again for the interest in the application of SAS to standard GP regression. Additionally to what we said in the main response, we must say that SAS is an approach which initially took place in the (probabilistic) representation learning scenario, and we consequently wrote a submission which summarises our work and discoveries on this particular area. We also agree that presenting SAS as a general learning approximation for both *supervised* and *unsupervised* GP models will be of higher impact for the community, but it would require a very thorough comparison with *all* baselines based on variational sparse methods, for instance, as Bauer et. al (2016) did. However, this would be a very different paper, perhaps out of the scope of our previous investigation and difficult to fit in the current manuscript. We hope that the reviewer will agree on this point.
> > >
> > > - *M. Bauer, M. van der Wilk, C. E. Rasmussen. Understanding Probabilistic Sparse Gaussian Process Approximations, NIPS, 2016.*

---

> ### Author Response · Authors · 2022-08-02
> **Response to Reviewer YAoZ -- (Part 2/3)**
>
>
> **Point 2**
> > *The claims in lines 29-34 are not well-supported. It would be much better to list the problems of training with inducing points (with references) instead of quotations.*
>
> Thanks for pointing this out. Just for clarification, we wanted to say that while variational learning of inducing points has progressed the supervised Gaussian process research immensely, learning inducing points in the unsupervised scenario of the GP-LVM remains challenging.
>
> First, in the supervised scenario (where the inputs are fixed), the inducing points are *not completely trivial to optimise, and often tricks [...] are required* (Bauer et. al 2016). This is further supported by Dutordoir (2020) who points out that the inducing points acts locally only and hence a large number may be required to cover the input space. This effect is even more pronounced in high-dimensional data (e.g. images) due to the curse of dimensionality.
>
> In the unsupervised setting, we hypothesize that this is further complicated as we have not only to optimize the inducing variables, but the inputs as well, since they are not fixed and observed anymore. In practice, this makes the optimization process even harder, as our early experiments showed us.
>
> **Point 3 / Questions**
> > *What is $K_{NN}$ in (1)?*
>
> Good catch. We did not write that in the paper. $K_{NN}$ is the kernel evaluated on a data point $x_n$ so $K_{NN} = k_{\theta}(x_n,x_n)$. We have included an explicit definition of $K_{NN}$ in the updated version.
>
> **Point 4 / Questions**
> > *Shouldn't $S_{CV}(x|R)$ in (3) be also conditioned on $z$?*
>
> Yes, it should as the scoring function for the GP-LVM is $p(x|z)$. For the Bayesian GP-LVM it should not be conditioned on $z$ as the scoring function here is $p(x)$.
>
> **Point 5 / Questions**
> > *How about including the formal definitions of $S_{PCV}$ and $S_{CCV}$ in the main paper instead of supplements?*
>
> Fair point. We will update entire Section 3 (on stochastic active sets) as this is confusing. Unfortunately, we would prefer to do this if the paper goes on for a final version under your supervision, especially considering your points 4-8.
>
> **Point 6 / Questions**
> > *$r$ appears on the middle term in (4) but only $R$ appears on the right hand side.*
>
> Right, this was not clear. We have addressed this as part of our answer to **Point 1** but to briefly summarize. Fong and Holmes (2020) shows the relation between the log marginal and the exhaustive cross validation score.  We can split this sum into two sums at a choice of $R$.
> $$
> \log p(x) = \sum_{r=1}^N S_{CV}(x;r) = \sum_{r=1}^R S_{CV}(x;r) + \sum_{r=R+1}^n S_{CV}(x;r)
> $$
> where the l.h.s. term corresponds to $S_{CCV}(x;R)$ and the r.h.s. one to $S_{PCV}(x;R)$
> This holds for all choices of $1 \leq R < N$. This also gives the definitions of $S_{CCV}$ and $S_{PVC}$ which both include the summation and for this reason only the limit of the sum is needed. We will update the paper with this which also addresses your **Point 5**.
>
>
> **Point 7 / Questions**
> > *The first equation in line 114 involves $R$ on the left but not on the right. If I understood correctly, $A=N-R$ (hence $R$ impicitly appears on the right. In that case, I would suggest using modifying the notation a bit to reflect this.*
>
> $R$ appears implicitly on the right hand side. We agree that the notation is somewhat confusing. We will update it according to the discussions here with the major update of Section 3 in the future.
>
>
> **Point 8 / Questions**
> > *Are $A$ and $A_p$ used interchangeably?*
>
> In short, the answer is no, these are not used interchangeably but this distinction is subtle and we realise that the definitions could be improved. We clarify the difference here.
>
> There are numerous way to pick $R$ points out of $N$, so we defined the set of all possible active sets as $\mathcal{A} = \mathcal{A}_1,\mathcal{A}_2,$ ... ,$\mathcal{A}_p$ ... ,$\mathcal{A}_C$, where $p$ indexes the permutation out of $\mathcal{C} = \binom{N}{A}$ possible permuations. Likewise, we define the set of all possible rest sets as $\mathcal{R} = \{\mathcal{R}_1,\mathcal{R}_2,\ldots,\mathcal{R}_p,\ldots,\mathcal{R}_C \}$ where $p$ also indexes the specific permutation out of $\mathcal{C}$.
>
>
> **Point 9 / Experiments**
> > *The loss curves in Figure 2 oscillate a lot.*
>
> This is a valid point. These curves show the loss *per batch*. We have updated the entire Figure 2, including all subplots, to show the loss *per epoch* which vastly reduces the noise in the curves.
>
> **Point 10 / Experiments**
> > *The MNIST latent space with Bayesian SAS is missing.*
>
> Indeed, you are right that the MNIST latent space for Bayesian is missing in Figure 3 which shows the data manifold with images. We included this one with points coloured according to class label in Figure 4. We showed the latent space in two ways because we think that the data manifold (e.g. the tilt of digits) and the colours (clustering by class) convey different messages.

---

> ### Author Response · Authors · 2022-08-02
> **Response to Reviewer YAoZ -- (Part 1/3)**
>
> Thanks for the very constructive feedback. We appreciate the time you clearly took to review our paper. We have addressed all your comments individually below. If any concerns remain, we would be also happy to clarify.
>
> **Point 1 / Weaknesses**
> > *I do not understand how to obtain (5) from (4), and how $S_{PCV}$ in (5) has $x_A$ on the left hand side of the conditioning while (3) has $x_A$ on the right.*
>
> Good point. We skipped some steps in the main manuscript, so we will also include what is derived below these lines in the appendix if the paper goes on. Essentially, we do an approximation of the GP log-marginal likelihood based on the link with cross validation (CV) made in Fong and Holmes (2020).
>
> Starting from the GP-LVM log-marginal likelihood, we first divide the observations into an active set $x_A$ and a rest set $x_R$, applying conditionals
> $$
> \log p(x|z) = \log p(x_R,x_A) = \log p(x_R|x_A,z) + \log p(x_A|z),
> $$
> without any loss of generality. Next, we assume that the *rest* set factorise over observations which diagonalises the covariance matrix in the first term. In the second term, only the latent points corresponding to the active set makes a difference, so we can condition on $z_A$ rather than all of $z$
>
> $$
> \log p(x|z) = \sum_{n} \log p(x_n|x_A,z) + \log p(x_A|z_A).
> $$
> Because these are jointly Gaussian distributions, we know how the conditional distributions are, as well and we can get these by regular Gaussian conditionals (e.g. Bishop, 2006, eq B.42-B.51 or Eq. 22 in Rasmussen and Williams, 2006) which is what our equation (5) states
> $$
> p(x_n|x_A,z) = \mathcal{N}(x_n|m_{n|\mathcal{A}},c_{n|\mathcal{A}})
> \text{ and }
> p(x_A|z_A) = \mathcal{N}(x_A|0,K_{\mathcal{A}\mathcal{A}}).
> $$
> The first term is a conditional distribution which we can interpret as the probability of some test data giving a training set (which Rasmussen and Williams (2006) also do) and the second term can be interpreted as the prior on $x_A$. Hopefully, this clarifies the second part of your question. If not, we are happy to discuss it.
>
> Next, we should look to Fong and Holmes (2020). On this, we will answer how to think about Equation 3. Fong and Holmes consider exhaustive leave-$R$-out cross validation which is the exact complement to choosing an active set $A$ as $N=A+R$. The exhaustive leave-$R$-out cross validation is given by Fong and Holmes (2020) in their Equation 5, where $s$ is a *scoring function* (i.e. the log-posterior predictive) on $t$-th of the possible $N$ choose $R$ possible test dataset $x_j^{(t)}$ conditioned on the corresponding training data $x_{1:N-R}^{(t)}$. We note that the training data $x_{1:N-R}^{(t)} = x_{1:A}^{(t)} = x_A^{(t)}$. The first sum accounts for all possible combinations of hold-out sets given $R$ (which is exactly symmetry to all possible combinations of active sets). The second sum is the factorisation over datapoints.
>
> In the GP-LVM scenario, the log-posterior predictive distribution is $p(x_n|x_A,z)$ for $n \in \mathcal{R}p$ and inserting that and rearranging the sums yields
> $$
> S_{CV}(x_{1:n};R) = \frac{1}{R} \sum_{n\in \mathcal{R}p} \frac{1}{C} \sum_{t=1}^C p(x_n|x_A,z)
> =  \frac{1}{R} \sum_{n\in \mathcal{R}p} \mathbb{E}_\mathcal{A}[p(x_n|x_A, z)]
> $$
> where $C = \binom{N}{R}$ and the expectation is taken of all possible active sets and therefor $\mathcal{A}$ rather than $\mathcal{A}_p$.

---

### Official Review · Reviewer_QibS · 2022-07-12

**Rating:** 5
**Confidence:** 4
**Soundness:** 2 fair
**Presentation:** 1 poor
**Contribution:** 2 fair

**Summary:**

The paper proposes a computationally efficient way to train the GP-LVM based on a connection between marginal likelihood and cross-validation in Fong and Holmes (2020). To make the estimation practical, stochastic estimation is used for sampling the size of the hold-out set and also the size of the active set. Amortization is also used for approximating $p(z|x)$. In the experiments, the performance of the method is compared with GP-LVM and VAE.

**Questions:**

1. How is R picked? Also, will the uniform sampling of r results in high variance of the estimate?
2. Why does Bayesian GP-LVM perform much worse than the proposed method or VAE? Some explanation and ablation studies would be helpful.

**Limitations:**

N.A.

**Strengths And Weaknesses:**

Strengths:

- Applying the idea of estimating MLL with cross-validation scores to GP-LVM is interesting.
- Several ablation studies on different active set sizes.

Weaknesses:

- It is not clear how different hold-out set sizes (R) will affect the proposed estimation. Also, I have difficulty understanding what “by first uniformly sampling R” means. Does it mean first pick R and then uniformly sample $r \in [1:R]$? I could not infer this from the pseudo-algorithm or experiments as well. If the sampling is by uniformly sampling $r \in [1:R]$, it seems necessary to have more discussion about whether the sampling is of high variance and if there are ways to reduce the variance.
- The presentation of the experimental results can be improved. Some important experiment details are deferred to the appendix, such as what kernel, NN architecture is used. They serve as very important information to understand the results. Also, not sure why RMSE/MAE is used as a metric as the MNIST or CIFAR10 are classification tasks. For evaluating the quality of the representation learned, it is not clear why 2 is selected as the number of dimensions. It might be better to consider trying a range of different dimensions.
- There are two major components in the proposed method: the SAS approximation and amortization. It would be better if there is some ablation study on both SAS and amortization. For example, by comparing GP-LVM-SAS-Amortization with GP-LVM-InducingPoint-Amortization and GP-LVM-SAS with GP-LVM-InudcingPoint without amortization.
- It seems that the baseline methods are only compared with the proposed method in terms of representation quality. There is no comparison on NLPD/RMSE/MAE. And it would be better to include some regression tasks as well.

---

> ### Author Response · Authors · 2022-08-02
> **Response to Reviewer QibS -- (Part 3/3)**
>
> **Question 2**
> > *Why does Bayesian GP-LVM perform much worse than the proposed method or VAE? Some explanation and ablation studies would be helpful.*
>
> Assuming that you refer to results in Table 2, we can explain the performance of the Bayesian GP-LVM by looking at Figure 4 which shows the learnt latent spaces for both MNIST and FashionMNIST. The Bayesian GP-LVM does not cluster as discriminatively as the SAS and the VAE and this is reflected in the classification accuracies.
>
> This naturally raises the question of why does the Bayesian GP-LVM seems to learn less structured latent spaces than the two other methods. In short, we suspect that this is related to the Bayesian GP-LVM being hard to train due to one optimizes both latent and inducing variables while they interact.
>
>
> This is also the driving motivation for exploring the alternative approach to scale up the GP-LVM. We can also say that Bayesian GP-LVM (like most GP-LVMs) is *very* sensitive to initial conditions. While the amortization neural network in SAS plays a key role, it also provides an initial randomness to the Bayesian GP-LVM that it seems the model cannot really recover from.

---

> > ### Comment · Reviewer_QibS · 2022-08-08
> > **Re: Response to Reviewer QibS**
> >
> > Thanks for the clarifications. It clears a lot of the confusion I was having when I was reading the manuscript. I would hope to point out that given GP-LVM is very sensitive to initialization, it might make sense to perform ablation studies on amortization and SAS by comparing methods over multiple initializations (like RL).

---

> > > ### Author Response · Authors · 2022-08-09
> > > **Extra reponse to Reviewer QibS**
> > >
> > > Thanks to the reviewer for the acknowledgement to our response. We are glad that a lot of the confusion on our manuscript is now clarified. Thus, we hope that this leads to a better consideration of our paper, including an increased score, since no unaddressed concerns remain.
> > >
> > > We also appreciate the points indicated by the reviewer on the *ablation*, *amortization* and *initialization* of experiments. We believe that these points have been considered independently in several parts of the results. We invite the reviewer to check out the new figures included in the updated appendix which also address some of the points mentioned. In addition, we would be also happy to restructure the experiments and include the details asked if the paper moves forward.

---

> > > > ### Comment · Reviewer_QibS · 2022-08-09
> > > > **Re: Extra reponse to Reviewer QibS**
> > > >
> > > > Thank you for the response. Similar to reviewer YAoZ, I think the paper can improve on its presentation for the methodology section and experiment figures. In Section 3, the introduction of SAS, especially around (3) and (4), is very hard to understand for me as well. Restructuring (starting from (4)) and adding more details would be very helpful in improving the paper's presentation. Your responses provide a lot more clarity to the methodology and I will increase my score based on this.
> > > > As a minor point, you might want to consider a related work section that talk about conditional marginal likelihood, variational inference and inducing points for GP-LVM.

---

> ### Author Response · Authors · 2022-08-02
> **Response to Reviewer QibS -- (Part 2/3)**
>
> **Point 4**
> > *For evaluating the quality of the representation learned, it is not clear why 2 is selected as the number of dimensions. It might be better to consider trying a range of different dimensions.*
>
> Good point. We have included two extra experiments with three and four latent dimensions.  Figures can be accessed in the Appendix, which we also updated. We would also like to elaborate our reasons for the choice of two latent dimensions: i) Generally, two latent dimensions is the *standard* visualization in sparse GP-LVMs (Titsias,2010; Gal,2015) and we were interested on being comparable to previous research. ii) The main focus of this work was to highlight the *richness* of the trained latent space relative to the baselines. If we had used more latent dimensions, we would have had to use visualization tools and we did not want that as a confounder when comparing the models visually.
>
> - *M. K. Titsias and N. D. Lawrence. Bayesian Gaussian Process Latent Variable Model, AISTATS, 2010.*
> - Y. Gal et al. Latent Gaussian Processes for Distribution Estimation of Multivariate Categorical Data, ICML, 2015
>
>
> **Point 5**
> > *There are two major components in the proposed method: the SAS approximation and amortization. It would be better if there is some ablation study on both SAS and amortization. For example, by comparing GP-LVM-SAS-Amortization with GP-LVM-InducingPoint-Amortization and GP-LVM-SAS with GP-LVM-InudcingPoint without amortization.*
>
> We appreciate the positive comment and it is indeed right that our proposed method consists of stochastic active sets and amortization.
>
> The model you call GP-LVM-InducingPoint-Amortization is the same as the model we refer to as Bayesian GP-LVM. As mentioned in line 227-229, we have enhanced this with an amortization network. We assume that the model you refer to is the Bayesian version of the GP-LVM-SAS-Amortization. These two model are compared in tables 1 and 2 and in figures 2, 3, and 4. Please correct us if we misunderstood your intention.
>
> As for the other proposed ablation study, the model you refer as GP-LVM-InudcingPoint without amortization is a regular Bayesian GP-LVM (Titsias and Lawrence, 2010). The other model you mention, the GP-LVM-SAS, we interpret as the Bayesian SAS GP decoder but excluding the amortization. It would be very interesting to disentangle the effect of SAS and the effect of the amortization but the effect of removing the amortization would be confounded by the intialisation as the performance of the non-amortized Bayesian GP-LVM crucially depends on the initalisation and therefore, the proposed ablation study would not be comparable to our experiments, simply due to difference in the initialisaitons. Essentially, we would unintentionally be comparing different initial conditions rather than evaluating the effect of amortization.
>
>
>
> **Point 6**
> >It seems that the baseline methods are only compared with the proposed method in terms of representation quality. There is no comparison on NLPD/RMSE/MAE. And it would be better to include some regression tasks as well.
>
> We compared the learnt representations both qualitatively and quantitively for baselines and our approach. For the quantitative comparison, we used the accuracy of a down-stream classifier (see Table 2) for both MNIST and Fashion MNIST. This is a standard method for comparisons in representation learning (rather than NLPD/RMSE/MAE).
>
> Sorry, we are somewhat confused as to what is referred with regression tasks. The GPLVM is essentially an unsupervised regression task and all experiments are focused on regression (though we used a down-stream classifer as an evaluation metric).
>
> **Question 1**
> > *How is R picked? Also, will the uniform sampling of r results in high variance of the estimate?*
>
> Good question. The selection of $R$ (which is equivalent as choosing the active set size $A$) is a similar problem as selecting $M$, the number of inducing points, for variational sparse GPs. Interestingly, this is a choice of learning quality, and the practitioner should think what is better in the tradeoff between computational cost and approximation. Similarly as happens with $M$ in sparse GPs, the larger $A$ is, the least we are approximating the log-marginal likelihood of the GPLVM. Experiments illustrated in Figure 2, Figure 5 and Table 1 also respond to the question about how the size of $A$, and $R$ affects to the performance and variance of the approximation.

---

> ### Author Response · Authors · 2022-08-02
> **Response to Reviewer QibS -- (Part 1/3)**
>
> We thank the reviewer for the useful comments and all the relevant feedback provided. We have addressed *all* your comments individually in the lines below. Particularly, we are interested in improving the clarity of the work, if there is still something unclear, as well as we provided extra experiments according to your points and others from the rest of reviewers.
>
> **Point 1A**
> > *It is not clear how different hold-out set sizes $R$ will affect the proposed estimation.*
>
> The hold-out set size $R$ can be interpreted as the size of the test set in cross validation (CV) and is related to $A$ by $N=A+R$. From a theoretical point of view, Vehtari & Ojanen (2012) states that *"Selecting a larger R has the interpretation of penalizing complexity, as complex models will tend to overfit to a small training set."*  On the other hand, a smaller $R$ (i.e. a greater than $A$) results in a better approximation as we use a larger full covariance matrix. This is exactly equivalent to the choice of inducing points in the Bayesian GP-LVM.
>
> - *Vehtari, A. and Ojanen, J. A survey of Bayesian predictive methods for model assessment, selection and comparison. Statistics Surveys, 6:142–228, 2012.*
>
>
> **Point 1B**
> > *Also, I have difficulty understanding what "by first uniformly sampling R" means. Does it mean first pick $R$ and then uniformly sample $r \in [1:R]$? I could not infer this from the pseudo-algorithm or experiments as well. If the sampling is by uniformly sampling $r \in [1:R]$, it seems necessary to have more discussion about whether the sampling is of high variance and if there are ways to reduce the variance.*
>
> We understand the confusion which is partly on our side. There are two things in it: 1) what we should do to obtain an unbaised estimator, and 2) what is practical when aiming to reduce the computational cost.
>
> As a reply to 1), to obtain an *unbiased* estimator, one should sample $R$ from the range $[1,\ldots,N]$. Next, one can sample the indices that determine which observations are selected for the active set and which are selected for the rest set. However, sampling $R$ from the range $[1,\ldots,N]$ means that we could sample $R=N$. This corresponds to an exact GP and has complexity $\mathcal{O}(N^3)$ so no speed is gained.
>
> For 2), in practise, we first pick $R$ (just like the number of inducing points is picked in inducing point methods). Next, for a batch $x$, we do the following:
> ```
> permutated_indices = torch.randperm(x.size()[0]) # Creates a uniform sampling of indices
> a_p = permutated_indices[:active_set_size]  # Indexing set for the active set
> rest = permutated_indices[active_set_size:]
> x_a = x[a_p] # Active set
> x_rest = x[rest] # Rest set
> ```
> We hope this clarified what was not clear from the paper. Additionally, error metrics in *Table 1* could help to see the effect of varying the active set size $A$ and hence, $R$.
>
>
>
> **Point 2**
> > *The presentation of the experimental results can be improved. Some important experiment details are deferred to the appendix, such as what kernel, NN architecture is used. They serve as very important information to understand the results.*
>
> We have updated the description of the experiments to include the mentioned experimental details and a reference to the elaboration in the appendix to the main paper. Thanks for pointing this out.
>
> **Point 3**
> > *Also, not sure why RMSE/MAE is used as a metric as the MNIST or CIFAR10 are classification tasks.*
>
> Perhaps, we have not been completely clear in our communication of experiments. In the first line of section 5.1, we indicated that we used an *unsupervised version of MNIST, FMNIST and CIFAR-10 (...).* While it is true that MNIST and CIFAR10 datasets are traditionally used for classification tasks, here we use them in an unsupervised setting so the training does not involve the class label. The main reason behind is that our work is focused in the unsupervised scenario, where we aim to do probabilistic representation learning using the GPLVM.
>
> The *class* labels of these datasets are solely used for plotting latent points in different colours to demonstrate the structure. For this reason, RMSE and MAE are suitable metrics and commonly used for GPLVM tasks, e.g. Lalchand (2022) and Bui (2015).
>
> Next, we wanted to quantify the learnt structure of the latent space and we did this using a down-stream classifier. We trained a nearest neighbor classifier on the test set encoded to the learnt representation. These results uses the label information and are shown in table 2. We hope this clarified the confusion and we have also clarifed the description in the paper.
>
> - *Lalchand et al. Generalised Gaussian Process Latent Variable Models (GPLVM) with Stochastic Variational Inference, AISTATS, 2022.*
> - *Bui and Turner, Stochastic Variational Inference for Gaussian Process Latent Variable Models using Back Constraints, Black Box Learning and Inference Workshop @ NIPS, 2015.*

---

### Author Response · Authors · 2022-08-02
**Response to Reviews**

We thank all reviewers for their useful comments, positive considerations and relevant feedback on our paper. It seems that the reviews are positive in general and acknowledges our main contributions and novelties, which we appreciate. We have addressed each comment and question individually below and we are glad to engage in discussion in case of more questions or concerns exist. We also updated both the manuscript and the appendix with the new results and experiments that reviewers asked for. To see what are the differences with the previous submission, we colored the updated parts. This is just temporal, and will be modified in the future, if needed.

---

### Meta-Review · Area_Chair_WzbU · 2022-08-26

**Recommendation:** Accept
**Confidence:** Certain

**Metareview:**

This paper proposes a stochastic algorithm to make inference in Gaussian process latent variables models more efficient. The key idea is to exploit an equivalence between the marginal likelihood and the leave-R-out cross validation score (due to Fong and Holmes) to reduce the original cubic complexity in the full dimensional space to cubic complexity on subsampled data plus the (linear) complexity of combining these estimates. Some reviewers felt the fundamental idea of the paper might be extendable to other GP models, and there were concerns about the clarity in several parts. (Though all agreed this was improved in the revision, further improvement remains possible.) There were also some concerns that the proposed idea presented a relatively small methodological leap. However, in the end the consensus was that the paper provides a valuable contribution linking intuitions about subsampling to formal guarantees and for this reason makes a valuable contribution.

**Award:**

No

---

### Decision · Program_Chairs · 2022-09-14

Accept